# REDUCING SPURIOUS CORRELATIONS IN CNNS VIA ATTENTION-BASED FEATURE AGGREGATION

## ABSTRACT

Convolutional Neural Networks (CNNs) often exploit spurious correlations in datasets, learning features that are superficially predictive but not causally relevant to the task, leading to poor generalization and fairness issues. The recently proposed Deep Feature Reweighting (DFR) technique aims to reduce the reliance of a trained model on spurious correlations by retraining its classification head on a target dataset, achieving state-of-the-art performance on various benchmarks. However, we find that DFR operates on entangled features, which limits its ability to simultaneously extract the core features while removing the influence of spurious features. We further find that this entanglement primarily occurs in the commonly used Global Average Pooling (GAP) layer, which indiscriminately collapses spatially distinct core and spurious features into a single representation. To overcome this limitation, we propose Deep Attention Reweighting (DAR), an attention-based aggregation module to replace GAP, which is retrained alongside the classification head. This attention mechanism allows for the adaptive weighting of spatial locations of the feature maps, enabling selective selection and suppression of core and spurious features at the feature map representation level before they are collapsed into entangled features. Across various metrics, datasets, and experimental settings, we empirically validate the effectiveness of DAR over DFR in resolving feature entanglement and mitigating spurious correlations.

## 1 INTRODUCTION

*Convolutional Neural Networks* (CNNs) have demonstrated exceptional performance in vision tasks but often fail to generalize in the presence of *spurious correlations*—patterns in the training data that happen to be correlated with the labels but unrelated to the underlying target function. When trained via *Empirical Risk Minimization (ERM)*, CNNs will indiscriminately absorb these correlations (Arjovsky et al., 2020; Nagarajan et al., 2020; Shah et al., 2020) to minimize the empirical risk. However, these patterns are simply shortcuts that do not consistently hold, resulting in a severe lack of robustness, inability to generalize, and unfairness (Geirhos et al., 2020). For instance, in the Waterbirds dataset (Welinder et al., 2010; Sagawa et al., 2019), the task of classifying birds as either water or land types is spuriously correlated with the water and land background features. A model that learns the spurious correlation might misclassify water birds with land backgrounds and land birds with water backgrounds.

*Deep Feature Reweighting (DFR)* (Kirichenko et al., 2023) is a simple yet effective post-hoc technique that retrains the last layer of an ERM-trained model on a small *balanced target dataset*, defined as one where there is a uniform distribution over classes within each spurious attribute such that the spurious attribute is no longer predictive of the label. This has achieved state-of-the-art performance on many popular benchmarks. The rationale is that DFR reduces the model's reliance on spurious features by adjusting the feature importance during retraining. However, we find its effectiveness limited due to the *entanglement* of the core and spurious features, where a single output feature encodes information for both. This entanglement forces DFR to navigate a tradeoff: retaining a feature to preserve core information while also including spurious correlations or suppressing a feature to remove spurious influences at the cost of losing the core information. As a result, DFR struggles to selectively suppress spurious features without inadvertently affecting core information. We further identified that this entanglement primarily occurs in the *Global Average Pooling* (GAP) layer (Lin et al., 2014), a widely used feature aggregation method in many popular CNN architectures

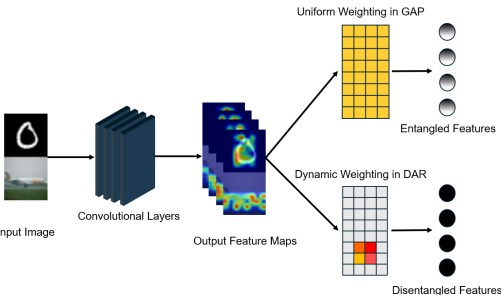

Figure 1: Illustration of GAP VS DAR. The input image from the Dominoes dataset consists of the spurious MNIST image concatenated with the core CIFAR image. After the feature extraction by the convolutional layers, we find that the output feature maps are entangled, where each feature map activates for both the core and spurious features at distinct spatial locations. GAP uniformly averages these feature maps to produce entangled output features, whereas DAR's attention mechanism extracts a disentangled core feature representation via dynamic spatial weighting.

(Szegedy et al., 2014; He et al., 2015; Huang et al., 2018; Sandler et al., 2019; Tan & Le, 2020) that uniformly averages each feature map across spatial locations into a single feature.

To overcome this limitation, we propose *Deep Attention Reweighting* (DAR), a novel approach that replaces the GAP layer with an attention-based aggregation mechanism that is retrained alongside the last layer. Unlike GAP, DAR learns the attention mechanism to adaptively identify and focus on the spatial locations in the feature maps that correspond to core features. This allows DAR to perform targeted feature selection and reweighting at the feature map representation level, where the core and spurious features are spatially disentangled, prior to the collapse into a feature vector. This difference between GAP and DAR is illustrated in Figure 1.

The key contributions of this work are as follows:

- **Feature entanglement in CNNs: Root cause and impact on DFR.**
  We introduce the Core Effect Percentage (CEP), Core Activation Percentage (CAP), and Core GradCAM Percentage (CGP) as quantitative measures of feature entanglement, and complement them with GradCAM visualizations as a qualitative diagnostic. Using these metrics, we experimentally show that feature maps are channel-entangled yet spatially separable: core and spurious features activate within the same feature map but at different locations. However, the GAP layer collapses this spatial separability into an entangled feature vector. Consequently, the post-GAP DFR reweighting method cannot reliably retain core information while removing spurious features.

- **Attention-based feature aggregation for disentanglement.**
  We introduce DAR, a novel, lightweight attention-based feature aggregation mechanism that assigns adaptive importance weights to spatial locations in feature maps, enabling the selective suppression of spurious features before they are collapsed into the final feature representation. We experimentally show that DAR significantly mitigates feature entanglement by enabling more effective isolation of core features.

- **Comprehensive empirical validation across benchmarks and setups.**
  We evaluate DAR on three spurious correlation benchmark datasets—Dominoes, Spawrious, and Spuco—against competitive baselines. Our results show that DAR consistently outperforms the baselines across various experimental configurations, demonstrating its robustness in mitigating spurious correlations.

## 2 RELATED WORKS

### 2.1 SPURIOUS CORRELATION

The challenge of spurious correlation in machine learning is studied under various frameworks, including the *correlation-versus-causation dilemma* (Lake et al., 2016; Lopez-Paz, 2016; Marcus, 2018), *invariant learning* (Peters et al., 2015; Heinze-Deml et al., 2018; Arjovsky et al., 2020),

*subpopulation shift* (Chen et al., 2023; Liang et al., 2023; Yang et al., 2023), and *group robustness* (Sagawa et al., 2019; LaBonte et al., 2023; Qiu et al., 2023), among others (see also (Ye et al., 2024) for a recent survey). The methods developed to address these challenges can be broadly categorized based on their dataset requirements. Firstly, methods that rely solely on the training dataset without additional information (Nam et al., 2020; Liu et al., 2021a; Pagliardini et al., 2022; Taghanaki et al., 2022; Zhang et al., 2022; Lee et al., 2023). Secondly, methods that require datasets from different environments (Ahuja et al., 2020; Arjovsky et al., 2020; Lin et al., 2022; Zhou et al., 2022). Thirdly, methods that require *group labels* (Sagawa et al., 2019), where each *group* is defined as a combination of the label and the spurious attribute (e.g., background in Waterbirds). Lastly, methods that require a small *target dataset* where spurious correlations are absent. Deep Feature Reweighting (DFR) (Kirichenko et al., 2023) falls under this category, retraining the final classification head of an ERM-trained model in a post-hoc manner on the target dataset, achieving state-of-the-art performance on several spurious correlation benchmarks.

This paper enhances the application of DFR on CNNs by identifying that its effectiveness is limited by the feature entanglement that occurs in the GAP layer. We propose an attention-based aggregation mechanism that replaces it, allowing the extraction of the core features at the feature map representation level before they are entangled with the spurious features.

## 2.2 DISENTANGLED REPRESENTATION LEARNING

*Disentangled Representation Learning (DRL)* aims to learn a feature representation where each feature independently captures a distinct factor of variation while being invariant to the others (Bengio et al., 2014; Locatello et al., 2020). A recent survey by Wang et al. (2024) highlights that Variational Autoencoders (Kingma & Welling, 2022) remain one of the most widely used frameworks, with many of the variants (Higgins et al., 2017; 2018; Dupont, 2018; Burgess et al., 2018; Kumar et al., 2018; Kim & Mnih, 2019; Kim et al., 2019; Mathieu et al., 2019; Chen et al., 2019) formulated under the assumption that the factors of variation are statistically independent. However, Träuble et al. (2021) found that when this assumption is violated, these methods fail to prevent entanglement and may even introduce a bias against disentanglement. This makes feature disentanglement particularly challenging in spurious correlations since the core and spurious features are statistically correlated.

Unlike DRL methods that seek to learn a disentangled representation, our method operates in a post-hoc setting where the model has already learned an entangled feature representation. Our paper identifies that the GAP layer is primarily responsible for this entanglement and replaces it with an attention mechanism that allows for the targeted extraction of task-relevant features.

## 2.3 ATTENTION-BASED ARCHITECTURES IN VISION

According to the survey by Han et al. (2023), the attention-based architectures in vision can be broadly categorized into two groups: pure attention architectures that replace convolutions entirely (Dosovitskiy et al., 2021; Touvron et al., 2021; Liu et al., 2021b; Wang et al., 2021b; Yuan et al., 2021) and hybrid architectures that integrate attention with convolutions (Wang et al., 2018; Woo et al., 2018; Hu et al., 2019; Bello et al., 2020; Wu et al., 2020). Generally, these architectures have fewer inductive biases and therefore rely on large-scale pre-training to learn these structures implicitly, but are able to overcome the limited receptive field of standard convolutions. Several works have further explored the robustness of these attention architectures to spurious correlations, finding that while they can focus on semantically meaningful regions, they remain susceptible to biases (Ghosal et al., 2022; Wang et al., 2021a; Yang et al., 2021; Yue et al., 2024).

In contrast to prior works that embed attention into the backbone architecture to model token interactions for feature representation learning, DAR retains the conventional CNN backbone, preserving its efficiency and inductive biases. Attention is applied as a lightweight, modular mechanism specifically designed for adaptive spatial reweighting in feature aggregation, to replace GAP.

Closest to ours is Jetley et al. (2018), which applies the simple dot-product attention to aggregate features across multiple layers. By contrast, our proposed architecture, designed for post-hoc adaptation, applies a more expressive multi-head, multi-layer attention mechanism after the final convolutional layer, where high-level, task-relevant features reside. Refer to Section 4.3 for the justification of the attention mechanism architectural design.

Table 1: CEP values (%) of the output prediction across various baseline methods. Refer to Section 3.2 for a detailed analysis.

| Method | $\text{ERM}_{\text{Core}}$ | $\text{ERM}_{\text{Train}}$ | DFR | $\text{DFR}_{\text{CNN}}$ | $\text{DFR}_{\text{FC}}$ |
|---|---|---|---|---|---|
| **CEP** | $95.6 \pm 0.1$ | $69.9 \pm 0.6$ | $81.6 \pm 0.8$ | $85.7 \pm 1.5$ | $83.0 \pm 1.2$ |

## 3 FEATURE ENTANGLEMENT AS A BOTTLENECK FOR POST-HOC REWEIGHTING

In this section, we examine the key limitation of Deep Feature Reweighting (DFR): its reliance on entangled ERM-learned features. Leveraging our Core Effect Percentage (CEP) metric to quantify feature entanglement, our analysis reveals the following: First, the output feature maps of convolutional layers are channel-entangled but spatially disentangled, with each map responding to both core and spurious image elements, but at different spatial locations. Second, the uniform averaging across spatial locations by the Global Average Pooling (GAP) layer collapses this spatial separability and yields output features that are entangled. Finally, DFR operates on these entangled features and is unable to eliminate reliance on spurious features.

### 3.1 QUANTIFYING FEATURE ENTANGLEMENT WITH CORE EFFECT PERCENTAGE

To quantify feature entanglement, we introduce the CEP metric, which measures the extent to which the core and spurious portions of the image influence the model's output features and predictions. This metric is derived from the *Controlled Direct Effect* in do-calculus (Pearl, 2012), similar to many interventional-based metrics (Carbonneau et al., 2024). Each input $\mathbf{x}$ is decomposed into the core $\mathbf{x}_{core}$ and the spurious $\mathbf{x}_{spu}$ components. The effect $E$ of each component on the model $f$ is quantified by the change in the model's output when that component is independently intervened on:

$$E_{\text{core}}(\mathbf{x}) = \left| f(\mathbf{x}) - f(\mathbf{x} \mid do(\mathbf{x}_{\text{core}} = \mathbf{x}'_{\text{core}})) \right| \tag{1}$$

$$E_{\text{spu}}(\mathbf{x}) = \left| f(\mathbf{x}) - f\left(\mathbf{x} \mid do(\mathbf{x}_{\text{spu}} = \mathbf{x}'_{\text{spu}})\right) \right| \tag{2}$$

Here, $\mathbf{x}'_{\text{core}}$ and $\mathbf{x}'_{\text{spu}}$ represent counterfactual replacements of the corresponding components, instantiated by substituting that component with the value from another datapoint while keeping the other component fixed. Finally, the CEP value is the expectation over inputs of the normalized core effect:

$$\text{CEP} = \mathbb{E}_{\mathbf{x}} \left[ \frac{E_{\text{core}}(\mathbf{x})}{E_{\text{core}}(\mathbf{x}) + E_{\text{spu}}(\mathbf{x})} \right] \times 100\%. \tag{3}$$

High CEP ($\approx 100\%$) indicates reliance on core features; low CEP ($\approx 0\%$) indicates reliance on spurious features; intermediate CEP ($\approx 50\%$) signifies reliance on both, i.e., entanglement.

### 3.2 EMPIRICAL ANALYSIS OF FEATURE ENTANGLEMENT

We run experiments on the MNIST-CIFAR Dominoes (Shah et al., 2020) dataset, which is constructed by concatenating MNIST (Deng, 2012) images with CIFAR-10 (Krizhevsky & Hinton, 2009) images. The labels correspond to CIFAR-10 classes (core features) but are spuriously correlated with the MNIST digits (spurious features). This dataset is well-suited for our study as it allows for controlled interventions on the input and straightforward interpretability through Grad-CAM (Selvaraju et al., 2019) visualization. Details regarding the experimental setup are available in Appendix A. We assessed feature entanglement across various baseline methods using the CEP metric at different levels of the model's representation: feature map (Figure 2), feature vector (Figure 3), and output prediction (Table 1).

$ERM_{Core}$ We train an ERM model on the dataset with a spurious strength of $0\%$, ensuring the model learns only core features since the spurious features are not predictive. As expected, CEP values are close to $100\%$ for all output levels: the feature maps (Figure 2a), features (Figure 3a), and predictions (Table 1).

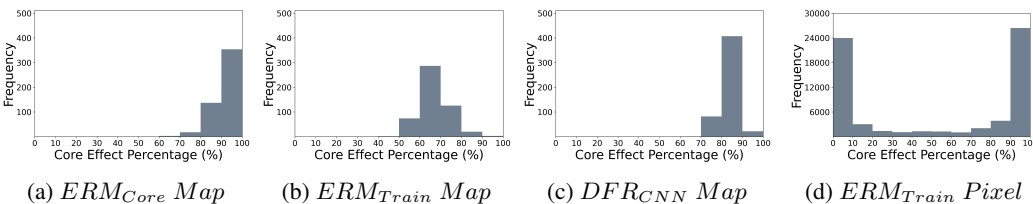

(a) $ERM_{Core}$ Map     (b) $ERM_{Train}$ Map     (c) $DFR_{CNN}$ Map     (d) $ERM_{Train}$ Pixel

Figure 2: Histogram of CEP values across 512 output feature maps for various baseline methods. Figures (a), (b), and (c) analyze the feature maps as a whole, whereas Figure (d) analyzes the feature maps at the pixel level. Refer to Section 3.2 for a detailed analysis.

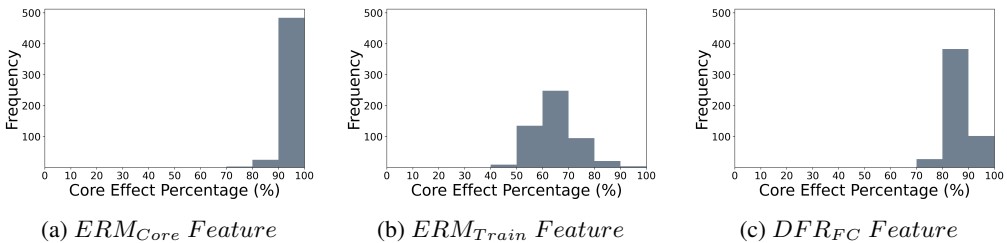

(a) $ERM_{Core}$ Feature     (b) $ERM_{Train}$ Feature     (c) $DFR_{FC}$ Feature

Figure 3: Histogram of CEP values across 512 output features for various baseline methods. Refer to Section 3.2 for a detailed analysis.

$ERM_{Train}$    When trained on the dataset with the spurious correlation, the ERM model learns both core and spurious features since both are predictive. The feature maps exhibit significant entanglement (CEP $\approx 50\%$ in Figure 2b). However, a more fine-grained pixel-level analysis reveals that different spatial locations within the same feature map often remain individually disentangled, with the pixel CEP values close to either $0\%$ or $100\%$ (Figure 2d). The subsequent GAP layer discards this spatial information by averaging over the entire feature map, collapsing otherwise disentangled spatial activations into entangled output features (Figure 3b). Finally, output predictions depend on both the core and spurious features, as reflected in the low CEP score of $69.9\%$ (Table 1).

$DFR$    DFR operates on the entangled ERM features (Figure 3b), reduces the reliance of the model's output prediction on the spurious features as seen from the improved CEP score to $81.6\%$ (Table 1), but is still insufficient to fully eliminate the influence of spurious features. To further investigate the claim that fast adaptation of a single layer is sufficient for feature disentanglement (Träuble et al., 2021), we evaluate two variants: $DFR_{CNN}$ and $DFR_{FC}$. $DFR_{CNN}$ additionally retrains the final CNN layer, leading to a slight improvement in CEP scores of the feature maps (Figure 2c). $DFR_{FC}$ introduces an additional fully connected layer to transform the output features, which slightly improves the CEP scores of the output features (Figure 3c). However, both variants remain insufficient to remove the influence of spurious features (CEP scores of $85.7\%$ and $83.0\%$ respectively in Table 1).

In addition to the CEP metric, we introduce the Core Activation Percentage (CAP) and Core Grad-CAM Percentage (CGP) in Appendix B to further quantify feature entanglement, along with qualitative inspections of the GradCAM visualizations in Appendix C; the results corroborate our findings.

## 4   DISENTANGLING FEATURES VIA DEEP ATTENTION REWEIGHTING

Building on our analysis in Section 3, we replace GAP with our proposed Deep Attention Reweighting (DAR), an attention-based aggregation mechanism that can adaptively assign importance to different spatial locations based on task relevance. This enables DAR to selectively suppress the spurious features at the feature map representation level, where they are spatially disentangled, before they are aggregated with the core features into an entangled representation. In this section,

we formally introduce our approach, outline the design considerations, and present experimental evidence demonstrating its effectiveness in reducing feature entanglement.

## 4.1 MOTIVATION: BEYOND GAP FOR FEATURE AGGREGATION

In CNNs, the GAP layer is widely used for feature aggregation by uniformly averaging each feature map across all spatial locations. However, as shown in Section 3, this uniform weighting indiscriminately aggregates both core and spurious features that are distributed across the spatial locations of the feature maps, resulting in feature entanglement that undermines post-hoc reweighting strategies. To address this limitation, we seek an aggregation mechanism that fulfills two specific criteria:

1. **Spatial weighting**: The aggregation mechanism should assign different importance values to different spatial locations within each feature map, allowing the model to emphasize task-relevant regions while suppressing irrelevant or spurious ones.

2. **Input-conditioned weighting**: The importance weights should be dynamically computed based on the input, allowing the model to adaptively focus on specific regions of that image.

## 4.2 ATTENTION-BASED FEATURE AGGREGATION

To satisfy the spatial weighting and input-conditioned weighting criteria outlined above, we propose replacing the GAP layer with an attention-based aggregation mechanism. Rather than uniformly averaging over all spatial positions, the model learns to assign input-dependent importance weights to different regions of the feature map.

Let $\mathbf{A}_i \in \mathbb{R}^{d \times H \times W}$ denote the activation map for input $\mathbf{x}_i$, where $d$ is the number of channels and $H \times W$ is the spatial resolution. The GAP operation aggregates each channel $j \in [d]$ across spatial locations to produce the output feature vector $\mathbf{h}_i \in \mathbb{R}^d$:

$$\mathbf{h}_i[j] = \sum_{h=1}^{H} \sum_{w=1}^{W} \frac{1}{H * W} \mathbf{A}_i[j, h, w] \tag{4}$$

We replace this with an attention-weighted aggregation:

$$\mathbf{h}_i[j] = \sum_{h=1}^{H} \sum_{w=1}^{W} \mathbf{a}_i[j, h, w] \mathbf{A}_i[j, h, w] \tag{5}$$

where $\mathbf{a}_i \in \mathbb{R}^{d \times H \times W}$ represents the attention weight for each spatial location for each feature map. This attention-based aggregation mechanism satisfies the two aggregation criteria introduced in Section 4.1: (1) spatial weighting via attention weights $\mathbf{a}_i[j, h, w]$, and (2) input conditioning, since the $\mathbf{a}_i[j, h, w]$ is dynamically computed based on each input feature map $\mathbf{A}_i$.

## 4.3 ARCHITECTURAL DESIGN OF ATTENTION MECHANISM

We adopt the scaled dot-product attention mechanism (Vaswani et al., 2017), with the following design considerations tailored for post-hoc retraining on a small target set, balancing expressiveness with overfitting risk:

1. **Query Vectors:** We use multiple learned query vectors, each of which independently computes attention weights over the spatial locations of the feature map, allowing DAR to capture a diverse set of distinct core-feature patterns.

2. **Positional Encoding:** We omit the use of positional encodings, which are typically added to inject information about the spatial position of each pixel. This preserves spatial invariance, ensuring that the attention weights depend only on the input content rather than their position.

3. **Attention Depth:** We employ a two-layer configuration, where the learned query vectors are first updated through a contextual transformation to align more closely with input-specific feature distributions before being utilized in the attention computation, yielding a more adaptive weighting mechanism.

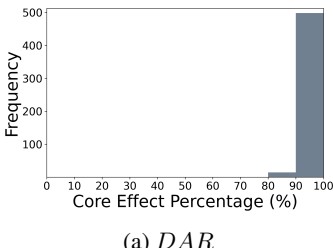

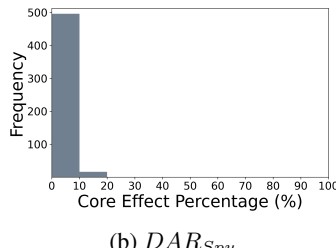

(a) $DAR$                                        (b) $DAR_{Spu}$

Figure 4: Histogram of CEP values across 512 output features for the proposed DAR method when predicting the core features (left) and the spurious features (right).

4. **Multi-Headed Attention:** We utilize a multi-headed attention setup, where each attention head independently learns a set of attention weights to focus on distinct spatial regions. This allows for a more expressive aggregation mechanism that can capture different features that are encoded across different feature maps at different spatial locations.

The full mathematical formulation of this attention architecture is provided in Appendix D.

### 4.4 DEEP ATTENTION REWEIGHTING ALGORITHM

The application of DAR involves three steps. First, train a base ERM model on the full training set. This yields a feature extractor that captures both the core and spurious features since they are both predictive of the labels. Second, replace the GAP layer with the attention mechanism. This replaces uniform pooling with selective aggregation at the feature map level, where the features are spatially disentangled. Finally, fine-tune the attention module and the classification head on a small, balanced target dataset where the spurious features are no longer predictive of the labels. This encourages the model to learn the targeted extraction of the core and suppression of the spurious features.

### 4.5 EXPERIMENTAL VALIDATION AND ANALYSIS

To assess whether DAR effectively mitigates feature entanglement, we apply the same diagnostic tools introduced in Section 3, using the same experimental setup detailed in Appendix A. As demonstrated in Figure 4a, the output feature vector from DAR predominantly captures the core features, resulting in model predictions that rely primarily on the core features, achieving an output prediction CEP score of $95.6 \pm 0.4\%$. To further examine the flexibility of the attention mechanism, we train $DAR_{Spu}$, a variant that predicts the spurious attributes instead (e.g., the MNIST digit in Dominoes). In this configuration, the output feature vector relies predominantly on spurious features (Figure 4b), with an output prediction CEP score of $6.1 \pm 0.7\%$. These results demonstrate the effectiveness of the attention-based feature aggregation architecture in selecting the spatial locations of the task-relevant features while ignoring the others. This is further supported by the additional entanglement metrics in Appendix B and GradCAM visualizations in Appendix C.

## 5 EXPERIMENTS

We evaluate our proposed DAR method across three benchmark spurious correlation datasets, under diverse experimental settings that vary the model architecture, spurious correlation strength, and pretraining conditions. Empirical results demonstrate that DAR consistently outperforms existing baselines. Our code is publicly available at https://anonymous.4open.science/r/DeepAttentionReweighting.

### 5.1 EXPERIMENTAL SETUP

We compare DAR against a set of widely used and competitive baseline methods for mitigating spurious correlations. Empirical Risk Minimization (ERM) is trained directly on the full dataset without additional information, serving as the naïve reference point. Both the Subsample and Group Distributionally Robust Optimization (gDRO) (Sagawa et al., 2019) methods assume access to *group labels* (information regarding the spurious attributes) and are commonly used as strong group-aware

references in the literature. The Subsample technique balances the training distribution by subsampling each group to ensure uniform distribution, thereby removing the statistical correlation between the spurious attributes and the labels. gDRO optimizes for the worst-group performance to ensure the model performs well across all groups, rather than just the majority. Deep Feature Reweighting (DFR) (Kirichenko et al., 2023) requires a small balanced target dataset for post-hoc reweighting and has achieved state-of-the-art performances. For fair comparison, the methods (ERM, Subsample, gDRO) that do not explicitly utilize the target dataset will have it shuffled into the training dataset. Overall, we find that these methods are the most relevant and competitive points of comparison, as they exploit strong additional information to mitigate spurious correlations (Idrissi et al., 2021), as opposed to recent methods that seek to relax this assumption (Ye et al., 2024).

The main experiment uses a randomly initialized ResNet18 model (He et al., 2015). This random initialization isolates each method's ability to learn the feature representation, independent of any pre-learned features from datasets like ImageNet. The methods are evaluated on three popular spurious correlation image datasets: MNIST-CIFAR Dominoes (Shah et al., 2020), Spuco-Animals (Joshi et al., 2023) (an extension of the Waterbirds dataset), and Spawrious (Lynch et al., 2023). We specifically selected vision benchmarks datasets with more than two classes, as prior works have shown that binary-class spurious correlation datasets can be too simple and may overestimate performance, in contrast to multi-class datasets, which provide a more challenging and realistic evaluation of robustness under spurious correlations (Idrissi et al., 2021). We additionally explore the performance of our proposed DAR method under three different experimental configurations: a larger ResNet-50 model, a weaker spurious correlation from 95% to 90%, and with ImageNet pre-training.

To evaluate the model's performance, we report both majority accuracy and minority-group accuracy, where the *group* is defined by the combination of the target label and a spurious attribute. In the datasets, the majority-group corresponds to the data points where the spurious correlation holds, whereas the minority-group corresponds to the data points where the spurious correlation does not hold. The experiments are conducted over three random seeds, reporting the mean and standard deviation. Additional details regarding the experimental setup can be found in Appendix A.

## 5.2 Main Results: Generalization Under Spurious Correlations

Table 2: Majority and Minority-Group Test Accuracy (%) Across Datasets

| Method | Dominoes | | Spawrious | | Spuco | |
|---|---|---|---|---|---|---|
| | Majority | Minority | Majority | Minority | Majority | Minority |
| ERM | $98.9 \pm 0.6$ | $69.2 \pm 2.6$ | $99.2 \pm 0.3$ | $93.9 \pm 0.6$ | $86.8 \pm 1.1$ | $45.4 \pm 1.7$ |
| Subsample | $74.5 \pm 2.7$ | $73.8 \pm 0.6$ | $93.5 \pm 1.9$ | $95.8 \pm 0.3$ | $50.7 \pm 1.9$ | $60.6 \pm 1.9$ |
| gDRO | $94.7 \pm 0.9$ | $75.4 \pm 1.3$ | $98.2 \pm 0.7$ | $96.1 \pm 0.7$ | $62.8 \pm 7.3$ | $58.2 \pm 1.5$ |
| DFR | $89.8 \pm 2.3$ | $78.3 \pm 0.8$ | $98.4 \pm 0.4$ | $96.3 \pm 0.6$ | $72.7 \pm 3.6$ | $63.3 \pm 0.8$ |
| DAR | $84.2 \pm 1.0$ | $\mathbf{79.8 \pm 1.9}$ | $96.6 \pm 0.8$ | $\mathbf{97.1 \pm 0.6}$ | $90.1 \pm 0.6$ | $\mathbf{66.5 \pm 2.8}$ |

Table 3: Minority-Group Test Accuracy (%) Across Experimental Configurations

| Method | Dominoes | | | Spawrious | | | Spuco | | |
|---|---|---|---|---|---|---|---|---|---|
| | ResNet-50 | 90% | Pretrained | ResNet-50 | 90% | Pretrained | ResNet-50 | 90% | Pretrained |
| ERM | $69.9 \pm 1.2$ | $76.2 \pm 0.8$ | $81.3 \pm 0.7$ | $91.0 \pm 1.0$ | $94.9 \pm 0.8$ | $96.0 \pm 0.6$ | $34.6 \pm 1.5$ | $56.8 \pm 0.2$ | $57.6 \pm 1.1$ |
| Subsample | $69.5 \pm 0.4$ | $77.5 \pm 0.8$ | $79.9 \pm 0.5$ | $94.6 \pm 1.2$ | $96.6 \pm 0.1$ | $\mathbf{99.0 \pm 0.2}$ | $61.9 \pm 4.0$ | $64.2 \pm 1.2$ | $74.5 \pm 1.4$ |
| gDRO | $72.8 \pm 1.2$ | $78.1 \pm 0.9$ | $85.6 \pm 1.2$ | $94.4 \pm 1.4$ | $96.9 \pm 0.8$ | $96.7 \pm 0.3$ | $60.7 \pm 1.7$ | $65.2 \pm 2.4$ | $63.2 \pm 1.3$ |
| DFR | $67.3 \pm 1.0$ | $82.2 \pm 0.3$ | $88.0 \pm 0.6$ | $96.4 \pm 0.3$ | $96.6 \pm 0.7$ | $97.2 \pm 0.2$ | $64.2 \pm 2.1$ | $68.3 \pm 1.2$ | $79.2 \pm 2.7$ |
| DAR | $\mathbf{78.2 \pm 0.1}$ | $\mathbf{83.6 \pm 0.2}$ | $\mathbf{88.1 \pm 0.2}$ | $\mathbf{97.1 \pm 0.8}$ | $\mathbf{97.3 \pm 0.4}$ | $98.0 \pm 0.3$ | $\mathbf{65.6 \pm 1.8}$ | $\mathbf{70.7 \pm 1.3}$ | $81.0 \pm 2.6$ |

First, ERM serves as a baseline and consistently exhibits the poorest performance across all datasets, as measured by the minority-group accuracy. Notably, the significant disparity in performance between the majority-group and minority-group accuracy (Table 2) highlights ERM's tendency to heavily rely on spurious correlations. This behavior is further illustrated in the training dynamics, as shown by the learning curves in Appendix E. This highlights that a high majority-group accuracy, and by extension, a high average accuracy, is extremely misleading since it can be achieved by learning the spurious correlation as opposed to the target function. This is consistent with the broader spurious correlation literature, where the minority-group accuracy is instead used as the accurate measure of the model's performance. Second, the Subsample and gDRO methods improve upon

ERM in terms of the minority-group accuracy. This underscores the importance of group-aware training objectives that explicitly leverage group label information to mitigate spurious correlations. Third, the DFR technique mostly outperforms the Subsample and gDRO methods, demonstrating its efficacy despite its simplicity and reliance only on a target dataset rather than full group annotations.

Across all datasets and experimental configurations (Table 2 & 3), our proposed DAR method consistently outperforms the DFR method and achieves the best performance across all methods. As discussed in Section 3, DFR's effectiveness is constrained by its dependence on entangled features. In contrast, Section 4 demonstrates how the attention-based aggregation mechanism in DAR facilitates a more effective extraction of the core features while suppressing spurious ones. This shift is directly reflected in the increased CEP compared to DFR, which provides a first-principles measure of feature reliance, confirming that its predictions are grounded in core rather than spurious signals. Such targeted feature selection ultimately translates into more robust and trustworthy performance, in addition to improvements in downstream accuracy. We note that, across datasets, DAR consistently outperforms DFR on minority accuracy. These gains can be further amplified by improving the quality of the learned representation during initial training via complementary robust feature-learning methods. We further discuss this in Section 6.

## 5.3 ABLATION RESULTS

### 5.3.1 ABLATION: DAR ARCHITECTURAL DESIGN

We examine two key DAR design decisions on Dominoes: (i) one-layer vs. two-layer attention and (ii) single-head vs. multi-head attention. The one-layer variant attains a minority-group accuracy of $75.7 \pm 1.5\%$, and the single-head variant reaches $76.6 \pm 0.4\%$. Both underperform our full two-layer, multi-head model, which achieves $79.8 \pm 1.9\%$. These results empirically support the architectural choices described in Section 4.3.

### 5.3.2 ABLATION: CONTROLLING FOR MODEL CAPACITY

To verify that DAR's performance gains do not simply arise from increased model capacity, we design two capacity-controlled baselines built on top of the DFR model. Starting from the original DFR architecture, we construct (i) $DFR_{FC}$, which fine-tunes an additional fully connected layer, and (ii) $DFR_{CNN}$, which fine-tunes an additional convolutional layer. Both variants introduce more trainable parameters than our DAR attention module and are trained under the same optimization and data settings. We observe that both capacity-controlled variants perform similarly to the original DFR model ($78.4 \pm 1.1$ for $DFR_{FC}$ and $77.3 \pm 2.3$ for $DFR_{CNN}$) and still underperform DAR ($79.8 \pm 1.9$). This indicates that simply increasing model capacity on top of DFR is insufficient to obtain DAR's robustness gains; instead, DAR's improvements stem from its targeted attention-based reweighting mechanism rather than from additional parameters.

### 5.3.3 ABLATION: INTEGRATING FEATURE LEARNING

As a post-hoc module applied to frozen ERM features, DAR may be limited by the quality of the learned representation. To study this, we retrain the backbone with a spectral decoupling loss (Pezeshki et al., 2020) designed to improve feature learning under spurious correlations, and apply DAR on top of these features. Under the Dominoes dataset, DAR improves from $79.8 \pm 1.9$ to $80.59 \pm 0.25$, highlighting the complementary nature of DAR with feature learning methods.

### 5.3.4 ABLATION: SPATIAL ENTANGLEMENT

To evaluate DAR in spurious-correlation settings without spatial separation between core and spurious features, we conduct experiments on the Colored-MNIST dataset (Arjovsky et al., 2020). Here, MNIST digits are colorized so that color is spuriously correlated with the digit labels, resulting in perfect spatial overlap of the spurious color feature and the core digit feature. DFR at-

tains $96.7 \pm 0.3\%$ minority-group accuracy, while our DAR achieves a very similar $96.9 \pm 0.5\%$. Combined with the results in Section 5.2, this clarifies DAR's operating principle: when core and spurious features are spatially disentangled, which is often the case in real-world datasets, DAR can leverage spatial structure to improve the performance; when they fully overlap, DAR does not introduce additional degradation and behaves similarly to DFR.

## 6 LIMITATIONS AND FUTURE WORK

First, DAR is a post-hoc technique that applies on top of the feature representation learned during initial training. An insufficient representation may cap DAR's performance despite substantial improvements in CEP. Indeed, DAR is complementary to approaches for robust feature learning (Pezeshki et al., 2020; Huang et al., 2020; Taghanaki et al., 2022); integrating such techniques to improve representation quality will further enhance DAR's performance.

Second, DAR, like most CNN models, can struggle under severe entanglement, where core and spurious features overlap within the same feature map and spatial location. Even so, DAR provides additional spatial dimensions for disentanglement relative to the commonly used GAP layer; this advantage underpins DAR's outperformance over GAP. Integrating existing disentangled representation learning techniques (Deng et al., 2022; Lee et al., 2021; Kong et al., 2022; Zhou et al., 2023) into backbone training can further mitigate such cases and improve the performance of the post-hoc DAR method.

## 7 CONCLUSION

We introduce Deep Attention Reweighting, an attention-based feature aggregation mechanism that replaces Global Average Pooling in CNNs. DAR addresses the issue of feature entanglement, a limitation of the post-hoc Deep Feature Reweighting method, by enabling spatially selective aggregation of feature maps. Through empirical evaluation across various metrics, datasets, and experimental setups, we demonstrate DAR's ability to mitigate feature entanglement and spurious correlations. More broadly, this work highlights how architectural choices can directly impact a model's ability to learn disentangled representations, which influences its ability to mitigate spurious correlations.

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

# A    EXPERIMENTAL SETUP

## A.1    DATASETS

**Dataset Description:**    The methods are evaluated on three spurious correlation image datasets: MNIST-CIFAR Dominoes (Shah et al., 2020), Spuco-Animals (Joshi et al., 2023), and Spawrious (Lynch et al., 2023). These benchmarks were chosen based on their widespread usage in spurious correlation robustness evaluation. Briefly:

- **Dominoes** is a synthetic dataset constructed by concatenating MNIST digits (spurious) with CIFAR-10 images (core), where the label corresponds to the CIFAR-10 object class. Unlike the binary setting in the original paper, we adopt the full 10-class classification task.
- **Spuco-Animals** extends the Waterbirds dataset to a more complex 4-class classification problem, where the task of identifying the animal type is spuriously correlated with the background.
- **Spawrious** contains AI-generated dog images where the task of identifying the dog breed is spuriously correlated with background style.

**Terminology:**    We adopt standard terminology from group robustness literature (Sagawa et al., 2019; Kirichenko et al., 2023):

- **Group:** The tuple $(y, a)$ of the target label $y$ and the spurious attribute $a$.
- **Majority Group:** The groups that constitute the largest percentage of the dataset, resulting in the spurious correlation between the attribute and the label. Consequently, the spurious correlation holds for examples in the majority group.
- **Minority Group:** The groups that constitute the smallest percentage of the dataset, with attributes and class labels that conflict with the spurious correlation.
- **Spurious Correlation Strength:** The probability that the spurious attribute $a$ is predictive of the label $y$. This is also the accuracy that the model would achieve if it simply learns the spurious correlations.
- **Minority Group Accuracy:** The average test accuracy computed only over the minority group samples, where the spurious correlation does not hold.
- **Majority Group Accuracy:** The average test accuracy computed only over the majority group samples, which is extremely misleading as the model can correctly classify them by simply learning the spurious correlations.

**Dataset Splits:**    The datasets are partitioned into four splits:

- **Training split:** used to learn the model's parameters. The sizes of this dataset for the Dominoes, Spuco-Animals, and Spawrious are 50000, 20000, and 24547, respectively.
- **Target split:** used by the DFR and DAR methods for post-hoc retraining. The other methods will have this shuffled into their training split. The size of this split is 1000 for all datasets.
- **Validation split:** used for hyperparameter tuning. The size of this split is 1000 for all datasets.
- **Test split:** used for the final model evaluation. The size of this split is 2000 for all datasets.

**Data Augmentation:**    The datapoints are normalized using dataset-specific statistics and augmented using Random Crop and Random Horizontal Flip.

## A.2    METHODS

In our experiments, we evaluate five methods: Empirical Risk Minimization (ERM), Subsample, Group Distributionally Robust Optimization (gDRO), Deep Feature Reweighting (DFR), and our proposed Deep Attention Reweighting (DAR). The ERM technique does not require any additional

information; the Subsample and gDRO techniques require group labels; the DFR and DAR techniques require an additional small target dataset.

We selected Subsample and gDRO as baselines due to their demonstrated effectiveness by leveraging group-specific information, outperforming methods that do not assume access to the group labels (Idrissi et al., 2021). Additionally, DFR is a key baseline, given its state-of-the-art performance. The implementation of the methods closely follows that of Idrissi et al. (2021) and Kirichenko et al. (2023).

## A.3 Training Setup

The main experiment uses a randomly initialised ResNet18 model (He et al., 2015). This random initialisation isolates each method's ability to learn the feature representation, independent of any pre-learned features from datasets like ImageNet. The models are trained for 200 epochs to ensure convergence. The models are optimised using the Adam optimiser (Kingma & Ba, 2015) with default beta values.

## A.4 Hyperparameter Tuning Setup

For the main experiment, we follow the hyperparameter-tuning setup described in (Yang et al., 2023). Specifically, we use the validation dataset to select the optimal hyperparameters for each method through grid search. These selected hyperparameters are then fixed and used to rerun the experiments under three random seeds, allowing us to report the average accuracy and its standard deviation. The three main hyperparameters we tune are: learning rate in $\{10^{-2}, 10^{-3}, 10^{-4}\}$, weight decay in $\{10^{-1}, 10^{-2}, 10^{-3}, 10^{-4}\}$, and batch size in $\{32, 64, 128\}$. The results reported are at the epoch with the highest minority group validation accuracy, i.e., early stopping. We do not perform additional hyperparameter tuning for the ablation experiments but instead use the best hyperparameters identified during the main experiments. The best hyperparameters for the different methods in the main experiment are summarized in Table 4.

Table 4: Best hyper-parameters found for the main experiment.

| Method | Dominoes | | | Spawrious | | | Spuco | | |
|---|---|---|---|---|---|---|---|---|---|
| | BS | WD | LR | BS | WD | LR | BS | WD | LR |
| **ERM** | 32 | $10^{-4}$ | $10^{-4}$ | 32 | $10^{-3}$ | $10^{-4}$ | 64 | $10^{-2}$ | $10^{-4}$ |
| **ERM$_{\text{train}}$** | 32 | $10^{-4}$ | $10^{-4}$ | 32 | $10^{-3}$ | $10^{-4}$ | 32 | $10^{-3}$ | $10^{-4}$ |
| **Subsample** | 32 | $10^{-4}$ | $10^{-3}$ | 32 | $10^{-4}$ | $10^{-4}$ | 128 | $10^{-3}$ | $10^{-3}$ |
| **gDRO** | 32 | $10^{-3}$ | $10^{-4}$ | 32 | $10^{-2}$ | $10^{-4}$ | 32 | $10^{-3}$ | $10^{-3}$ |

The hyperparameter tuning for DFR and DAR follows the setup in the original DFR paper (Kirichenko et al., 2023). Hyperparameter tuning is performed on the base model $ERM_{train}$, which only utilizes the training split.[1] This base model is then frozen for the retraining step. During this retraining step, both the DAR and DFR steps include an additional regularization parameter that is tuned. The best hyperparameter for this simple tuning step is not reported.

We do not perform hyper-parameter tuning for the ablation studies and simply use the best hyper-parameters found for each method in Table 4.

## A.5 Evaluation Setup

To evaluate the model's performance, we report both majority-group accuracy and minority-group accuracy. The majority-group accuracy is not a good evaluation of the model's performance since the model could achieve this by learning the spurious correlation instead of the target function, making it misleading. By extension, the average group accuracy is also not a good metric. Therefore, the minority group accuracy is used to benchmark the model's performance by evaluating its performance on the datapoints where the spurious correlation does not hold. This is consistent with the

---

[1]In contrast, $ERM$ uses both the training and target splits combined.

spurious correlation literature. All methods are repeated for three random seeds for robustness, with the mean and standard deviation reported.

## A.6 COMPUTATIONAL RESOURCES

Each experiment was run on a single Nvidia H100 GPU. The approximate GPU hours taken to run the different methods and datasets are reported in Table 5.

Table 5: Approximate GPU hours taken to run the methods.

| Method | Dominoes | | Spawrious | | Spuco | |
|---|---|---|---|---|---|---|
| | ResNet-18 | ResNet-50 | ResNet-18 | ResNet-50 | ResNet-18 | ResNet-50 |
| Subsample | 1 | 2 | 3 | 4 | 2.5 | 3 |
| ERM/gDRO/DFR | 5 | 9.5 | 15.5 | 21 | 13.5 | 16 |
| DAR | 6 | 11.5 | 22 | 29 | 20.5 | 24.5 |

The Subsample technique runs the fastest since the resulting dataset is much smaller after a significant amount of the majority group's data points are removed. The ERM, gDRO, and DFR techniques all take approximately the same time to run since none of them introduces significant overhead to ERM. The DAR takes slightly longer since it requires the retraining of the attention module. In contrast, the retraining of the classification head, a simple linear layer, in DFR can be performed efficiently.

## B SUPPLEMENTARY METRICS AND ANALYSIS FOR FEATURE ENTANGLEMENT

### B.1 CORE ACTIVATION PERCENTAGE (CAP) METRIC

Given an output feature map of the final convolutional layer, the activation in the top half corresponds to the MNIST input. In contrast, the activation at the bottom half of the feature map corresponds to the CIFAR input. We compute the *Core Activation Percentage (CAP)* for the $j$-th feature map as follows:

$$CAP_j = \mathbb{E}_i \left[ \frac{\sum_{h=H/2}^{H} \sum_w^W |\mathbf{A}_i[j, h, w]|}{\sum_{h=1}^{H} \sum_{w=1}^{W} |\mathbf{A}_i[j, h, w]|} \right] * 100\% \tag{6}$$

where $H$ and $W$ are the height and width of the feature map, and $\mathbf{A}_i \in \mathbb{R}^{d \times H \times W}$ is the activation of the feature map for input $\mathbf{x}_i$. The CAP metric is computed for all of the model's output feature maps.

### B.2 CORE GRADCAM PERCENTAGE (CGP) METRIC

We use the GradCAM (Selvaraju et al., 2019) technique to obtain a heatmap highlighting the parts of the input image the model focuses on when making its prediction. Similar to the CAP, the top half of the GradCAM heatmap corresponds to the MNIST input, while the bottom half corresponds to the CIFAR input. We measure the *Core GradCAM Percentage (CGP)* as follows:

$$CGP = \mathbb{E}_i \left[ \frac{\sum_{h=H/2}^{H} \sum_w^W |\mathbf{G}_i[h, w]|}{\sum_{h=1}^{H} \sum_{w=1}^{W} |\mathbf{G}_i[h, w]|} \right] * 100\% \tag{7}$$

where $H$ and $W$ is the height and width of the GradCAM heatmap, and $\mathbf{G}_i \in \mathbb{R}^{H \times W}$ is the Grad-CAM heatmap foarer input $\mathbf{x}_i$. The CGP metric is computed for the model's output.

### B.3 ABLATION RESULTS AND ANALYSIS

The results from these two additional ablation metrics (Table 6 and Figure 5) support the conclusions drawn in Section 4:

| Method | ERM$_{\text{Core}}$ | ERM$_{\text{Train}}$ | DFR | DFR$_{\text{FC}}$ | DFR$_{\text{CNN}}$ | DAR | DAR$_{\text{Spu}}$ |
|---|---|---|---|---|---|---|---|
| CGP | $91.0 \pm 0.3$ | $66.8 \pm 0.4$ | $70.0 \pm 1.5$ | $82.5 \pm 1.2$ | $79.6 \pm 1.8$ | $95.8 \pm 0.5$ | $2.2 \pm 0.5$ |

Table 6: Core GradCAM Percentage for Model's Output Prediction.

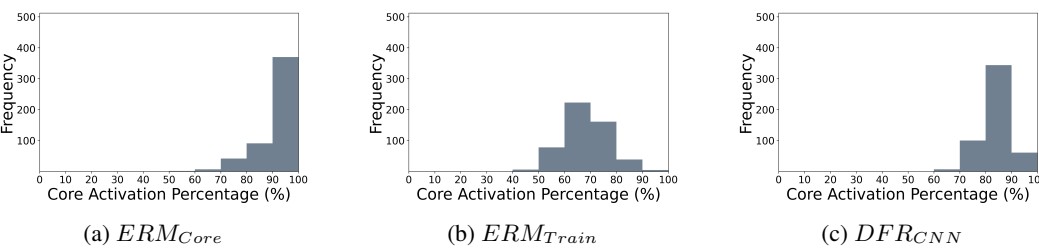

(a) $ERM_{Core}$        (b) $ERM_{Train}$        (c) $DFR_{CNN}$

Figure 5: Histogram of CAP values across 512 output feature maps.

1. $ERM_{core}$: When trained on a dataset where the spurious correlation is absent, the feature maps (Figure 5a) and output prediction ($CGP = 91.0\%$, Table 6) demonstrate a strong reliance on core features.

2. $ERM_{train}$: When trained on the dataset with spurious correlation, the feature maps (Figure 5b) exhibit significant entanglement between core and spurious features, resulting in an output that depends on both ($CGP = 66.8\%$, Table 6).

3. $DFR$: While DFR improves the CGP score ($CGP = 70.0\%$, Table 6) compared to $ERM_{train}$, it is still insufficient to fully eliminate the influence of spurious features as it still relies on the entangled feature maps (Figure 5b).

4. $DFR_{CNN}$: Modifying $DFR$ to include the retraining of the final CNN layer reduces reliance on the spurious features in both the feature maps (Figure 5c) and the output prediction ($CGP = 79.6\%$, Table 6), but remains insufficient to fully remove their influence.

5. $DFR_{FC}$: Modifying $DFR$ to retrain a two-layer classification head (instead of the original single-layer) reduces the reliance on the spurious feature in the output prediction ($CGP = 82.5\%$, Table 6), but remains insufficient to fully remove their influence.

6. $DAR$: The proposed $DAR$ method significantly outperforms $DFR$, effectively extracting core features while almost completely eliminating spurious feature influence ($CGP = 95.8\%$, Table 6). Alternatively, it can be configured to extract only the spurious features ($CGP = 2.2\%$, Table 6).

## C GRADCAM IMAGES

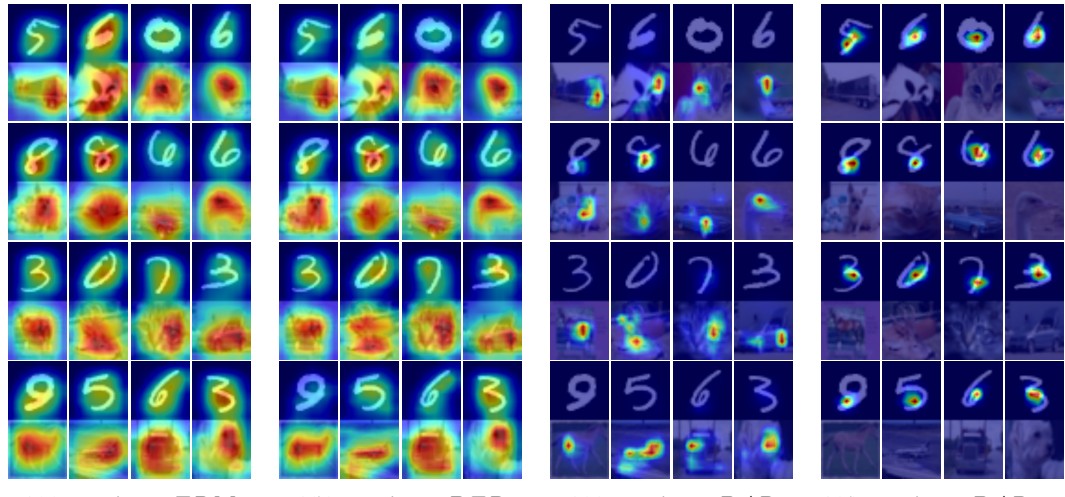

1(a) Dominoes $ERM$       1(b) Dominoes $DFR$       1(c) Dominoes $DAR$       1(d) Dominoes $DAR_{Spu}$

Figure 6: A random sample of 16 GradCAM images from the test datasets for $ERM$, $DFR$, $DAR$, $DAR_{Spu}$ models that were obtained from the main experiments for the Dominoes dataset.

Through visual inspection of Figure 6, we observe that the ERM model relies on both the core and spurious features. The DFR model reduces the reliance on the spurious features, but it is insufficient to completely remove its influence due to the entanglement of features. Finally, our proposed DAR method can almost completely eliminate the influence of these spurious features, focusing almost exclusively on the core features. Alternatively, it can predict the spurious attributes and focus almost exclusively on the spurious features. We also note that the attention mechanism learns to assign attention weights that are dynamic and extremely focused, with the attention for each input image focusing solely on the most discriminative regions.

# D  MATHEMATICAL FORMULATION OF DAR

In Section 4.3, we introduced the architectural design used in our proposed DAR model. In this appendix, we mathematically introduce the architecture, which is based on the scaled dot-product attention mechanism:

$$SDPA(\mathbf{Q}, \mathbf{K}, \mathbf{V}) = softmax\left(\frac{\mathbf{Q}\mathbf{K}^T}{\sqrt{d}}\right)\mathbf{V} \tag{8}$$

where $\mathbf{Q}, \mathbf{K}, \mathbf{V}$ represent the query, key, and value matrices, respectively, while $d$ represents the dimension of the vectors. The multihead attention architecture extends the $SDPA$ as follows:

$$head_t = SDPA(\mathbf{Q}\mathbf{W}_Q^t, \mathbf{K}\mathbf{W}_K^t, \mathbf{V}\mathbf{W}_V^t), \quad t = 1, \dots, n_{\text{head}} \tag{9}$$

$$MHA(\mathbf{Q}, \mathbf{K}, \mathbf{V}) = Concat(head_1, ..., head_{n_{head}})\mathbf{W}_O \tag{10}$$

where $n_{head}$ refers to the number of heads and $\mathbf{W}_Q^t, \mathbf{W}_K^t, \mathbf{W}_V^t \in R^{d, d_{head}}$ are linear projection matrices that map from the original dimension $d$ to the dimension of each head $d_{head} = d/n_{head}$. $\mathbf{W}_O \in R^{d,d}$ is the output projection matrix.

Using these attention modules, we can define our two-layer attention architecture as:

$$\mathbf{Q}' = MHA(\mathbf{Q}, \mathbf{A}, \mathbf{A}) \tag{11}$$

$$\mathbf{H} = MHA(\mathbf{Q}', \mathbf{A}, \mathbf{A}) \tag{12}$$

where $\mathbf{Q} \in \mathbb{R}^{k \times d}$ represents $k$ learnable query vectors, $\mathbf{A} \in \mathbb{R}^{HW \times d}$ represents the output activation map which is used as both the key and value matrices. In this formulation, the first layer updates the query vectors in the context of the feature map, before using the updated queries to extract the core features.

Finally, $f^{attention}$ represents the feature embedding extracted from the feature map $\mathbf{A}$ based on the learnable query vectors $\mathbf{Q}$. We take the average of these feature vectors to obtain the final feature representation:

$$h = \frac{1}{k}\sum_{i=1}^{k}\mathbf{H}[i] \in \mathbb{R}^d \tag{13}$$

# E  TRAINING PLOTS

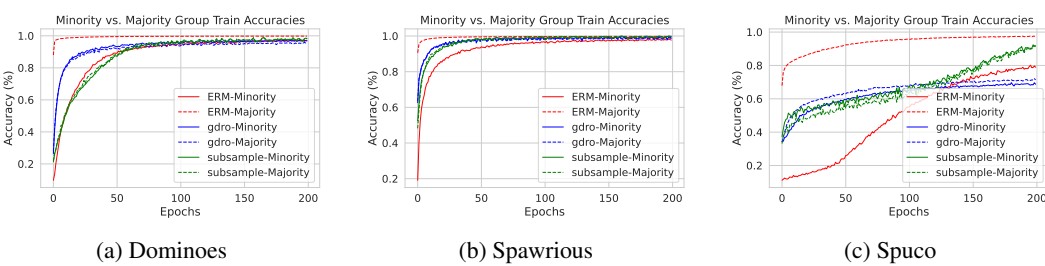

(a) Dominoes        (b) Spawrious        (c) Spuco

Figure 7: Learning curves of the Dominoes, Spawrious and Spuco datasets, plotted for the ERM, subsample, and gDRO methods. The plots consist of the minority-group and majority-group training accuracies.

For ERM, we observe that at the start of the model training, there is an immediate spike in the model's training accuracies for the majority-group accuracies, whereas the training accuracies for the minority-group accuracies drop below random guessing. This is a clear indication that the model has learned the spurious correlations, allowing it to easily classify the data points from the majority-group. These spurious correlations, however, do not generalize to the data points from the minority-group. Over time, we see that the performance for the minority-group data points gradually increases, indicating that the model eventually learns the core features to correctly classify these data points. However, there is still a huge gap between the majority-group and the minority-group test-accuracies (Table 2, which indicates that the model still heavily relies on the spurious features. This additional shows how the learning of the spurious features allows the model to easily correctly classify the examples from the majority group, making the evaluation on these data points not reflective of the model's true performance.

In contrast, for the subsample and gDRO methods, we observe that the performance of the majority-group and the minority-group generally increases at the same rate. This implies that the group information helps the model to learn the core correlations and ignore the spurious correlations that only perform well for certain groups.

