# OpenReview forum: "Reducing Spurious Correlations in CNNs via Attention-Based Feature Aggregation"
_ICLR.cc/2026/Conference — Submitted to ICLR 2026_

### Official Review · Reviewer_nj1D · 2025-10-30

**Soundness:** 3
**Presentation:** 3
**Contribution:** 3
**Rating:** 8
**Confidence:** 4

**Summary:**

This paper identifies that the effectiveness of the state-of-the-art Deep Feature Reweighting (DFR) method for reducing spurious correlations in CNNs is limited by feature entanglement. The authors pinpoint the Global Average Pooling (GAP) layer as the primary cause, as it indiscriminately averages spatially distinct core and spurious features into a single, entangled representation. To address this, they propose Deep Attention Reweighting (DAR), a novel post-hoc approach that replaces the GAP layer with a lightweight attention mechanism. This module is retrained to adaptively weight spatial locations within feature maps, enabling it to selectively aggregate task-relevant core features while suppressing spurious ones before they are collapsed. Empirical results demonstrate that DAR significantly improves feature disentanglement and consistently outperforms DFR and other baselines, particularly on minority-group data where spurious correlations are absent.

**Strengths:**

The core originality of this paper lies not in inventing a new attention mechanism. While attention has been used for feature aggregation before, the authors are the first to explicitly link the architectural choice of the Global Average Pooling (GAP) layer to the limited performance of the post-hoc debiasing technique, DFR. This causal connection that GAP creates the very feature entanglement DFR struggles with, is a novel and valuable insight. Furthermore, the introduction of the Core Effect Percentage (CEP) metric, grounded in interventional do-calculus, provides a principled and original way to quantify this entanglement, moving beyond qualitative observations to concrete measurement.
The proposed method, Deep Attention Reweighting (DAR), is a logical and well-motivated solution to the diagnosed problem. The authors construct a clear, linear narrative: DFR is limited by entanglement, GAP causes entanglement, our attention-based DAR solves it. This finding encourages the community to think more critically about the inductive biases embedded in standard architectures and their downstream effects on fairness and generalization.

**Weaknesses:**

* The central premise of DAR is that core and spurious features, while channel-entangled, are spatially separable in the final feature maps. The method's success hinges on this assumption. However, the paper does not explore scenarios where this assumption might be violated. For instance, a spurious correlation could manifest as a subtle, global texture or style change that co-locates with the core object. In such cases, a spatial attention mechanism would have no distinct regions to up-weight or down-weight, and its advantage over GAP would likely diminish.  Have you considered scenarios of spatial entanglement? for instance, where a spurious feature is a global texture or a style applied to the entire image rather than a localized object?
* As a post-hoc method, DAR's performance is capped by the quality of the feature representations learned by the initial ERM-trained backbone. If the backbone fails to learn spatially separable features to begin with, DAR will have little to work with. The paper acknowledges this as a limitation and area for future work.
* The paper details a specific multi-head, two-layer attention architecture but does not provide an ablation study to justify these choices over simpler alternatives. It is unclear if all this complexity is necessary or if a simpler dot-product attention (as in Jetley et al., 2018) would have sufficed. Could you provide a brief ablation or justification for the multi-head, two-layer design over a simpler attention mechanism (e.g., single-head or single-layer)?

**Questions:**

See weaknesses.

---

> ### Author Response · Authors · 2025-11-21
>
> We thank the reviewer for the valuable and thoughtful feedback and now respond to the comments in detail below.
>
> 1) Does DAR still provide an advantage when core and spurious features are spatially entangled?
>
> We acknowledge that DAR exploits the spatial separation of features when it is present, but does not provide additional gains when they are spatially entangled, as shown in **our newly added ablation experiment** in Section 5.3.4. We’d also like to refer the reviewer to our first response to Reviewer NSbP (Comment 1) for further details.
>
> 2) Is DAR’s effectiveness limited by the quality and spatial separability of the features learned by the initial ERM-trained backbone, given that it is a post-hoc method operating on a fixed representation?
>
> We agree with the reviewer that, as a post-hoc method, DAR is inherently limited by the quality and spatial separability of the features learned by the initial ERM-trained backbone, and is therefore complementary to approaches that improve the underlying representation. To explore this, **we conducted a new experiment** where we trained the backbone with a spectral decoupling loss designed to improve robustness under spurious correlations and then applied DAR on top of these features. As reported in our newly added analysis in Section 5.3.3, this further improves DAR’s performance, supporting the view that DAR benefits from—and can be combined with—stronger representation learning methods.
>
> 3) Ablation on the proposed DAR architecture.
>
> We thank the reviewer for this suggestion. In response, **we added a new ablation** in Section 5.3.1 that benchmarks our attention module against single-head and single-layer variants. Our proposed architecture outperforms these simpler alternatives, providing empirical support for the design choices and motivation outlined in Section 4.3.

---

> > ### Comment · Reviewer_nj1D · 2025-11-26
> >
> > I appreciate the authors’ thorough and comprehensive response and their effort to include additional experiments.

---

### Official Review · Reviewer_w4eU · 2025-10-30

**Soundness:** 2
**Presentation:** 3
**Contribution:** 1
**Rating:** 2
**Confidence:** 4

**Summary:**

This paper addresses the problem of CNNs learning spurious correlations, which harms their ability to generalize and which can also lead to fairness issues. While the existing Deep Feature Reweighting (DFR) technique attempts to mitigate this by retraining the classification head, the authors find its effectiveness is limited by feature entanglement. They identify the Global Average Pooling (GAP) layer as the primary cause of this entanglement, as it harmfully combines spatially distinct core which are causally relevant and spurious which are non-relevant into a single representation. To overcome this, the paper proposes Deep Attention Reweighting (DAR), a novel approach that replaces the GAP layer with an attention-based aggregation module. This attention mechanism is retrained alongside the final layer, learning to adaptively assign importance weights to different spatial locations within the feature maps. This process allows DAR to selectively focus on core features and suppress spurious ones before they are collapsed into an entangled feature vector. The authors introduce new metrics to quantify entanglement and demonstrate through comprehensive experiments on three benchmark datasets that DAR significantly outperforms DFR and other baselines in resolving feature entanglement and reducing reliance on spurious correlations.

**Strengths:**

The authors clearly and effectively situate their work by building upon the existing DFR method, identifying a precise and plausible limitation: feature entanglement. They convincingly argue that the Global Average Pooling layer is a primary culprit, which provides a strong and logical motivation for their proposed solution. The core idea of Deep Attention Reweighting is interesting and intuitive. Instead of trying to fix entangled features after they've been combined, DAR intervenes at the source, using an attention mechanism to selectively aggregate features before they are collapsed, which is a straightforward approach to the problem.

Furthermore, the proposed method has the advantage of being relatively simple and practical. The concept of replacing the GAP layer with a lightweight attention module—and only retraining this new module alongside the classification head—makes it an accessible/efficient solution. It doesn't require a complex re-architecture or a full, costly retraining of the entire model. Another methodological strength is the authors' contribution of three novel metrics (CEP, CAP, and CGP); these are well-motivated and specifically designed to quantify the abstract concept of feature entanglement, allowing for a much more rigorous and appropriate evaluation of their method's success beyond just standard accuracy scores.

**Weaknesses:**

The main weaknesses of the paper lie in the great imbalance between the well-motivated problematic and its unconvincing validation. Given that the proposed DAR method is relatively simple and straightforward, I believe that it is up to the authors to demonstrate its superiority to DFR as well as other methods, which the paper fails to deliver. The experimental section is rather sparse, lacking the extensive benchmarks and in-depth qualitative analyses needed to be really convincing; in fact, one of the very few qualitative analysis is relegated to the appendix. The lack of thorough validation makes it difficult to assess the true weight of the paper's contribution. The paper's structure and writing also seems a bit weird and distracting from its main message. The introduction, for example, is written in a confusing manner, which very early dives into specifics of a commonly used dataset in spurious correlations (Waterbirds) in its second opening sentence.. Instead, it should first establish the general problem, the limitations of existing solutions, and a clear, high-level motivation for the proposed method.

The paper also suffers from presentation issues and sections that could be significantly more concise and shorter. Figures, such as Figure 1 and Figure 4, mostly take up excessive space without being proportionally informative; this space could have been better utilized for more extensive experiments (both qualitative and quantitative). On a smaller/less important note, the histograms in Figure 2, would be more illustrative if they displayed density (percentages) rather than raw frequency. Furthermore, large sections of the text feel redundant. The lengthy paragraph analyzing DFR (lines 252-261), while perhaps accurate and appropriate for the appendix, appears to offer no novel insights and could be significantly shortened. Similarly, the entire section on the architectural design of the attention mechanism (Section 3.4) could be condensed into a single paragraph, with the specific technical details moved to the appendix to improve readability and focus.

Apart from presentation, the paper contains critical methodological ambiguities and unsubstantiated claims. A key component of the DAR algorithm is the use of a "small balanced target dataset" (line 363), yet this concept is left not well-defined. The authors provide no specifics on the required proportion of this dataset, nor do they define what "balanced" means in the context of spurious correlations (e.g., in the Waterbirds dataset, what would this balanced mean? Or in the less specific case of a more general dataset?). The subsequent claim that the model "will correctly learn" on this dataset simply because spurious features are no longer predictive is stated as an obvious fact, but it would be much stronger if it was backed up by experimental evaluations. This lack of rigor extends to most of the results in the paper, where the empirical evidence is unconvincing. In Table 2, DAR's performance on majority accuracy is not as good, and its improvement over DFR on minority accuracy seems to often be statistically insignificant. While Table 3 shows stronger results, the overall evaluation is not elaborate enough to prove that DAR offers a meaningful and significant advantage over DFR, or other proposed methods.

**Questions:**

Please see weaknesses.

---

> ### Author Response · Authors · 2025-11-21
>
> We thank the reviewer for the thoughtful and constructive feedback and respond to the comments in detail below.
>
> 1) Improvements in paper writing and presentation:
>
> We sincerely thank the reviewer for the detailed comments on the writing. We have carefully incorporated this feedback and revised the main text to be more concise, while preserving the material we consider essential (please see the revisions highlighted in blue, in particular Figure 1, Section 3.2, and Section 4.3).
>
> 2) Strengthening the validation of the well-motivated problem:
>
> We are grateful that the reviewer found the paper to be well motivated. To address the concern about the balance between motivation and validation, we **(1) pair each motivated problem with its corresponding validation, and (2) introduce new validation experiments**, as detailed below.
>
> 2-1) One-to-one pairing between motivation and validation:
>
> Firstly, we observe that under standard ERM training, feature maps are entangled across channels (each map mixes core and spurious signals), which is undesirable for robustness, yet remain disentangled across spatial locations (core and spurious activations occur in different regions of the same map). We validate this by computing our proposed CEP metric at the feature-map level, where scores concentrate around 50% (Figure 2b), and at the pixel level, where values cluster near 0% or 100% (Figure 2d), indicating channel-level mixing but spatial separation. Motivated by this observation, DAR is designed to explicitly exploit this spatial disentanglement to recover a more robust core representation.
>
> Secondly, we find that the widely used GAP layer within CNN exacerbates entanglement in the output representation by collapsing each spatially structured feature map into a single scalar, thereby averaging over core and spurious regions indiscriminately. This is validated by the CEP scores for each output feature, which are close to 50% (Figure 3b). In contrast, our proposed DAR method exploits the spatial dimension using attention weights to selectively emphasize the spatial location of the core features, producing a disentangled core feature representation. This is validated by the CEP scores for each output feature, which are close to 100% (Figure 4a).
>
> Thirdly, we find that the existing spurious-correlation solution DFR that operates on the entangled feature representation cannot fully eliminate the reliance on the spurious features. This is validated by measuring the output prediction CEP score, which is 81.6% (Table 1). In contrast, our proposed DAR solution operates on a disentangled core feature representation and can better eliminate the reliance on the spurious features. This is validated by the output prediction CEP score of 95.6%.
>
> Finally, we find that the ability of our proposed DAR method to extract a disentangled core features representation translates to an improved model performance, as seen from the consistent improvements in minority-group accuracy across datasets (Dominoes, Spuco, Spawrious) and experimental settings (model architecture, pretraining status, strength of spurious correlation).
>
> 2-2) Additional experiments:
> To further validate the effectiveness of our proposed methods, **we conducted four additional experiments as follows**:
>
> (i) DAR across architectural designs (Section 5.3.1): We evaluate the one-layer and single-head architecture variants of DAR, providing additional validation for the design choice in Section 4.3.
>
> (iii) DAR over additional fine-tuning baselines (Section 5.3.2): We evaluate DAR against two DFR variants that introduce more trainable parameters than DAR, verifying that DAR’s performance gains do not simply arise from increased model capacity.
>
> (iii) DAR with feature learning baselines (Section 5.3.3): We evaluate DAR on top of an ERM-trained model using the spectral decoupling loss that is designed to improve feature learning under spurious correlations, finding that DAR is complementary to feature learning methods.
>
> (iv) DAR when the spatial separation assumption does not hold (Section 5.3.4): We evaluate DAR on a setting where core and spurious features are fully overlaid, showing that DAR does not introduce additional degradation and performs on par with existing methods when spatial separation is absent.
>
> 3) Use of minority-group performance:
>
> In our evaluation, we follow the widely adopted practice in the spurious-correlation benchmarks and use the minority-group accuracy as the primary metric of robustness against spurious correlation. Across the datasets, the spurious attribute is correlated with the labels for the majority-group, but not for the minority-group. A model that mistakenly learns the spurious correlation can therefore still achieve a high majority-group accuracy. Thus, how well the model has learned the core features must be measured on the minority-group, where the spurious correlation is not predictive.

---

> > ### Author Response · Authors · 2025-11-21
> >
> > 4) Clarification of “balanced” target dataset:
> >
> > Thank you for pointing this out. By “balanced,” we mean that the spurious attribute is no longer predictive of the label. For example, in Waterbirds, we use approximately equal numbers of samples for each (label, background) group (water/land bird × water/land background), so background alone cannot predict the bird label. More generally, this means an approximately uniform distribution over classes within each spurious group. We have updated the text in Section 1 to make this definition explicit.
> >
> > Training the model on such a balanced dataset reduces its reliance on the spurious features, as reported in [1]. We follow this setting in our experiments. We have revised the manuscript to remove the earlier phrasing that the model will “correctly learn” the core features and instead describe this more cautiously in terms of encouraging reliance on core rather than spurious features.
> >
> > [1] Polina Kirichenko, Pavel Izmailov, and Andrew Gordon Wilson. Last layer re-training is sufficient for robustness to spurious correlations. In The Eleventh International Conference on Learning Representations, 2023. URL https://openreview.net/forum?id=Zb6c8A-Fghk.

---

> > > ### Comment · Reviewer_w4eU · 2025-11-25
> > >
> > > I thank the authors for their detailed reply. Below are a few final comments:
> > >
> > > Regarding section 5.3.1, 5.3.2, 5.3.3 and 5.3.4, these analysis should be added in clearly formatted tables rather than in ablation paragraphs like this. What dataset is 5.3.2 conducted on, is it Dominoes?
> > >
> > > Regarding "balanced" target dataset: does this mean that you have modified the Waterbirds dataset? In my opinion this should be clarified and potentially written as one of the limitations of the approach then. For example, in the Waterbirds dataset, if one was to use the training data in a balanced fashion this would greatly reduce the amount of available data:
> > >
> > > - Landbirds on land: ~ 3,498
> > > - Landbirds on water: ~ 184
> > > - Waterbirds on water: ~ 1,057
> > > - Waterbirds on land: ~ 56
> > >
> > > Meaning you could only use approx. 200 samples out of the roughly 4000 in the data?
> > >
> > > Overall, I stand by my initial judgement, as the method is simple and straightforward it needs significantly more quantitative and qualitative evaluations. I refer the authors to the work by [1] in which many datasets and evaluation methods (both quantitative and qualitative) can be found.
> > >
> > > [1] Yang, Y., Zhang, H., Katabi, D. and Ghassemi, M., 2023. Change is hard: A closer look at subpopulation shift. arXiv preprint arXiv:2302.12254.

---

> ### Author Response · Authors · 2025-11-26
>
> We sincerely thank the reviewer for the comments and respond to the comments in detail below.
>
> 1. What dataset is 5.3.2 conducted on?
>
> We thank the reviewer for pointing out this ambiguity. All ablation experiments in Section 5.3, including those in Section 5.3.2, are conducted on the Dominoes dataset.
>
> 2. Limitations of the balanced target dataset:
>
> Firstly, we would like to clarify that the ERM backbone is trained on the full (imbalanced) Waterbirds training set, and that the group-balanced subset is used only in the post-hoc retraining step. Secondly, the small group-balanced subset is **not a limitation of the method**, but rather a constraint of the optimization problem that the post-hoc retraining algorithm is designed under, which is addressed by restricting the adaptation to a lightweight module to mitigate overfitting on this small target set. Finally, we note that there is emerging work [2] that aims to relax this assumption by using reweighted losses on the full target set instead of a strictly balanced subset, which stands orthogonal and complementary to our contribution.
>
> 3. Additional datasets and evaluation methods:
>
> The reviewer notes that [1] includes “many datasets and evaluation methods.” We agree that [1] provides a **broad benchmark for subpopulation shift**; however, our work presents an **architectural improvement**, DAR, that is specifically focused on handling **spurious correlations** in **image classification**.
>
> i) Regarding the choice of the datasets, although [1] considers 20 datasets, only 4 of them are image-based spurious correlation benchmarks that match the problem studied in our paper.
>
> ii) Regarding the evaluation metric, in line with the spurious-correlation literature, we adopt the minority-group accuracy (which in our setting is more robust than worst-group accuracy) as the primary robustness metric, which is pointed out by [1] as the gold-standard evaluation metric. Since our test sets are group-balanced, additional aggregate metrics such as precision, recall, F1, or balanced accuracy provide limited extra information.
>
> iii) Regarding the choice of baseline methods, [1] finds that DFR achieves the best performance, which is exactly why DFR, among others, is used as a competitive baseline comparison.
>
> The extensive experimental coverage in [1] reflects its goal as a **benchmarking paper**, where the expectation is a broad evaluation on a wide range of benchmarks. By contrast, our work is a **method-focused paper**, where the expectation is a targeted evaluation on representative benchmarks. In this sense, we concentrate our experiments on widely used spurious-correlation datasets (Dominoes, Spuco, Spawrious, CMNIST) and demonstrate our gains over SOTA DFR as well as most relevant spurious-correlation baselines (gDRO, subsampling), over various experimental setups (model architecture, pretraining status, spurious correlation strength), plus four new experiments added in the rebuttal phase, evaluating DAR’s architectural design choice, dependence on additional parameters introduced, compatibility with feature learning techniques, and its spatial separability assumption. Our experimental setup closely follows the foundational papers on spurious correlations [3-6], and we invite a closer comparison between our experimental setups and theirs. Finally, we would be grateful to know whether there are any particular quantitative or qualitative evaluations you consider essential for making a stronger case for acceptance.
>
> [1] Yang, Y., Zhang, H., Katabi, D. and Ghassemi, M., 2023. Change is hard: A closer look at subpopulation shift. arXiv preprint arXiv:2302.12254.
>
> [2] Shikai Qiu, Andres Potapczynski, Pavel Izmailov, and Andrew Gordon Wilson. Simple and fast group robustness by automatic feature reweighting. Proceedings of the 40th International Conference on Machine Learning, Jul 2023. URL https://proceedings.mlr.press/v202/qiu23c.html.
>
> [3] Polina Kirichenko, Pavel Izmailov, and Andrew Gordon Wilson. Last layer re-training is sufficient for robustness to spurious correlations. In The Eleventh International Conference on Learning Representations, 2023. URL https://openreview.net/forum?id=Zb6c8A-Fghk.
>
> [4] Shiori Sagawa, Pang Wei Koh, Tatsunori B. Hashimoto, and Percy Liang. Distributionally robust neural networks for group shifts: On the importance of regularization for worst-case generalization, 2019. URL https://arxiv.org/abs/1911.08731.
>
> [5] Evan Zheran Liu, Behzad Haghgoo, Annie S. Chen, Aditi Raghunathan, Pang Wei Koh, Shiori Sagawa, Percy Liang, and Chelsea Finn. Just train twice: Improving group robustness without training group information. CoRR, abs/2107.09044, 2021a. URL https://arxiv.org/abs/2107.09044.
>
> [6] Badr Youbi Idrissi, Martin Arjovsky, Mohammad Pezeshki, and David Lopez-Paz. Simple data balancing achieves competitive worst-group-accuracy, 2021. URL https://arxiv.org/abs/2110.14503.

---

### Official Review · Reviewer_NSbP · 2025-11-01

**Soundness:** 2
**Presentation:** 3
**Contribution:** 2
**Rating:** 2
**Confidence:** 4

**Summary:**

Convolution neural networks (CNNs) often exploit spurious correlations in datasets, leading to poor generalization and fairness issues. The recent proposed deep feature reweighting operates on the features entangled with core and spurious features because of the global average pooling (GAP) layer commonly used in CNNs. To address this, the paper proposes an attention-based aggregation module called deep attention reweighting (DAR) to replace GAP. This module allows dynamic selection and suppression of core and spurious features. Experiments across various metrics and datasets validate the effectiveness of DAR over DFR in mitigating spurious correlations.

**Strengths:**

- The paper proposes core effect percentage (CEP), core activation percentage (CAP), and Core GradCAM percentage (CGP) to quantify the entanglement between spurious and core features. These metrics can be useful for analyzing shortcut learning behaviors in models.

- The proposed attention-based aggregation module is effective in mitigating spurious correlations on three datasets.

- The paper is well-written and easy to follow.

**Weaknesses:**

- The motivation that spurious and core features in a feature map are spatially separated is only supported by  experiments on the MNIST-CIFAR Dominoes dataset where spurious and core features are spatially separated by design. While this dataset allows controlled interventions on the input and straightforward interpretability through Grad-CAM, it is unclear whether this motivation also holds for real-world datasets where spurious and core features are not well separated.

- In the attention-based feature aggregation defined in Eq. (5), the same attention weight is applied to different feature maps. It is possible that different feature maps have different distributions of spurious and core features. Using the same attention weights for different maps may again entangle spurious and core features. Are there any justifications on why using the same attention weight matrix for different feature maps?

- Lack of ablation studies: the introduction of an additional attention module adds more parameters for fine-tuning. It is unclear whether the performance gains are from additional parameters or from the proposed attention module. It would be better to also give the result of fine-tuning the last convolution layer along with the final classification layer and the results of fine-tuning additional architectures (e.g., fully connected non-linear layers) other than attention, to show that the performance gains are not achieved by increasing parameters.

- It is unclear whether the learned attention weights actually assign higher weights to core features than spurious features. Adding a visualization of the attention weights would be beneficial.

**Questions:**

- Would adding additional parameters (not with the proposed attention module) for fine-tuning increase performance?

- Does the claim that spurious and core features in a feature map are spatially separated generally hold in other datasets?

- Why do you use the same attention weights for different feature maps?

- Could you provide a worst-group accuracy comparison between DFR and DAR?

---

> ### Author Response · Authors · 2025-11-21
>
> We are grateful to the reviewer for the valuable feedback and now respond to the comments in detail below.
>
> 1) Does the assumption that spurious and core features are spatially separable in a feature map hold for real-world datasets?
>
> DAR benefits from spatial separation between core and spurious signals in the feature maps. This spatial separation is common in real-world spurious correlation conditions, for example, when the foreground object contains the core information, whereas the background surrounding context encodes the spurious correlation, as seen in the Waterbirds dataset. Because convolutional layers generally preserve spatial structure and locality due to the local connectivity inductive bias, features that are spatially separated in the input tend to remain distinguishable across spatial locations in intermediate feature maps. In the Dominoes dataset, this spatial separation is explicit and makes this effect measurable and visually interpretable. However, our other benchmark datasets are real-image datasets designed to study spurious correlations in more realistic settings where overlap between the spurious and core features can occur, and DAR still outperforms the commonly used GAP baselines, consistent with DAR’s design principle of mitigating feature entanglement via attention-based spatial reweighting.
>
> At the same time, we agree that strict spatial separation does not hold universally. In extreme regimes where the spurious attribute is perfectly overlaid on the core signal, there is no spatial separation in the input, and we do not expect DAR to provide additional gains. A **newly added ablation experiment** in Section 5.3.4 verifies this behavior and explicitly clarifies the role of spatial separation in DAR. We also emphasize that existing representation learning disentanglement techniques are complementary to DAR and can further enhance its effectiveness, as discussed in Section 6.
>
> 2) Why is it justified to use the same shared spatial attention weight matrix across all feature maps, as seen in Eq. (5)?
>
> Equation (5) indeed shows a single attention weight matrix being applied across all feature maps. This expression is written for clarity in the case of simple dot-product attention. However, as described in Section 4.3, our actual DAR module uses multi-head spatial attention, where each head has its own query/key projections and therefore learns a distinct spatial attention map. This design addresses the concern that different feature maps may contain different mixtures of core and spurious features at different spatial locations: different heads can specialize in different spatial patterns, rather than enforcing a single shared attention mask over all channels. We have revised Eq. (5) in the updated manuscript to explicitly reflect this multi-head formulation.
>
> To further support this design choice, **we conducted a new ablation experiment** (Section 5.3.1) comparing single-head and multi-head attention in DAR. We observe an improvement in performance with the multi-head variant over the single-head one, confirming that using multiple attention matrices—as correctly pointed out by the reviewer—is indeed beneficial in practice.
>
> 3) Are the performance gains truly due to the proposed attention module, rather than just the extra parameters it introduces?
>
> The performance gains primarily stem from the proposed attention module rather than merely from an increase in parameters. Section 4 provides a first-principles analysis of DAR, showing how its attention mechanism enables targeted extraction of core features. In addition, **we added a new ablation experiment** (Section 5.3.2) comparing DAR against two new baselines built on top of the SOTA baseline DFR, where we fine-tune additional fully connected and convolutional layers, respectively. Notably, these baselines have more trainable parameters than DAR, yet the new results show that DAR outperforms both. This supports our claim that the improvement is due to the attention-based aggregation and its ability to selectively focus on informative spatial locations, rather than simply having a larger model.

---

> > ### Author Response · Authors · 2025-11-27
> >
> > 4. Visualization of attention weights
> >
> > We agree that visualizations are important for understanding what the model attends to. While we do not directly plot the raw attention weight matrices, we provide **visualizations using Grad-CAM in Appendix C**, which reflect the attention maps learned by our model. These Grad-CAM heatmaps show that, compared to the ERM and DFR baselines, our method **consistently concentrates activation on the core object regions while suppressing spurious background cues**, thereby providing evidence that the learned attention focuses on core rather than spurious features.
> >
> > 5. Worst-group accuracy metric
> >
> > In this work, we intentionally adopt minority-group accuracy as our primary robustness metric rather than worst-group accuracy.  The minority-group is defined as the union of all subgroups for which the **spurious attribute is not predictive of the label**; this set precisely corresponds to the population on which reliance on spurious correlations is harmful, and is therefore the target population on which we evaluate the robustness to spurious correlations. In contrast, worst-group accuracy focuses on the single most challenging subgroup, which can be dominated by very small or atypical subgroups and can be **statistically unstable**, especially in datasets with many groups, such as the Dominoes dataset with 100 groups (10 labels x 10 spurious attributes). For these reasons, we believe minority-group accuracy is the more relevant and robust metric in our setting.

---

### Author Response · Authors · 2025-12-04
**Summary Comment**

This paper studies robustness to spurious correlations in CNNs by identifying a key limitation of the state-of-the-art post-hoc Deep Feature Reweighting (DFR) method: it operates on features that are entangled by the standard Global Average Pooling (GAP) layer, which averages spatially distinct core and spurious signals into a single representation. This paper proposes Deep Attention Reweighting (DAR), a simple and lightweight attention-based aggregation module that replaces GAP and is retrained together with the final classification head, enabling the model to disentangle the core and spurious features from the feature map representation level prior to spatial collapse. We introduce three intervention-based metrics to quantify feature entanglement and show that DAR improves the extraction of disentangled core features. Across several standard spurious correlation benchmarks and settings (different backbones, pretraining, and spurious strengths), DAR yields consistent gains in minority-group accuracy, demonstrating that the improved representation translates into better robust performance.

**Summary of key technical contributions of the paper, echoed by the reviewers:**
1) **Metrics for quantifying feature entanglement:**

This paper introduces three metrics to quantify feature entanglement: the Core Effect Percentage (CEP), Core Activation Percentage (CAP), and Core GradCAM Percentage (CGP). All three reviewers recognize the proposed metrics as a meaningful contribution. Reviewer nj1D emphasizes that CEP, grounded in interventional do-calculus, provides a **principled and original** way to quantify feature entanglement, while Reviewer w4eU highlights that the metrics are **well-motivated** and designed to capture this otherwise abstract notion. These metrics turn qualitative observations into **concrete quantitative measurements** (reviewer nj1D), support more **rigorous and appropriate evaluation** of methods beyond standard accuracy (reviewer w4eU), and serve as **useful tools for analyzing shortcut-learning behaviors** in models (reviewer NSbP).

2) **Diagnosis of DFR’s limitations:**

Our paper explores the feature representation at the feature map level, finding that they are channel-entangled while spatially-disentangled. The GAP layer collapses each channel across spatial locations into entangled features, which limits the performance of the post-hoc DFR. Reviewers nj1D and w4eU both find the **core argument of the identified limitation convincing**, providing an **explicit link from the architectural choice of the GAP layer to the limited performance of the post-hoc DFR technique**. Reviewer nj1D further emphasizes that this diagnosis is a **novel and valuable insight** into the performance of DFR, and it encourages a closer **examination of the inductive biases in standard architectures and their downstream impact on fairness and generalization**.

3) **Proposed DAR Solution:**

Our paper proposed DAR, an attention-based aggregation module that adaptively assigns importance to different spatial locations to selectively suppress the spurious features at the feature map representation level, where they are spatially disentangled, to extract a disentangled core feature representation. Reviewers nj1D and w4eU both view the proposed DAR solution as a **logical and well-motivated solution** to the diagnosed problem. Reviewer w4eU further highlights the advantage of it being a **simple, practical, and efficient solution**.

4) **Empirical effectiveness of proposed DAR:**

We provide experiments across several standard spurious correlation benchmarks and settings (different backbones, pretraining, and spurious strengths), finding that DAR yields consistent gains in minority-group accuracy over various baselines. Reviewer NSbP notes that, on the three evaluated datasets, the proposed attention-based aggregation module is **effective in mitigating spurious correlations**.

---

> ### Author Response · Authors · 2025-12-04
> **Summary Comment**
>
> **Key Criticisms and Rebuttal**
>
> Reviewer w4eU argues that there are insufficient experimental results, further pointing to the benchmark paper [1] as a reference for the datasets and evaluation metrics. In our rebuttal, we argue that the extensive experimental coverage in [1] reflects its goal as a benchmarking paper, where the expectation is a broad evaluation on a wide range of benchmarks. By contrast, our work is a method-focused paper, where the expectation is a **targeted evaluation on representative benchmarks**. This is what our paper achieves, concentrating our experiments on widely used spurious-correlation datasets (Dominoes, Spuco, Spawrious) and demonstrating our gains over SOTA DFR as well as most relevant spurious-correlation baselines (gDRO, subsampling), over various experimental setups (model architecture, pretraining status, spurious correlation strength). We also reiterate the key arguments of our paper and the corresponding experimental results that validate them. Finally, we added four new ablation studies to further validate DAR.
>
> 1. **Spatial separability assumption:**
>
> Reviewer NSbP and Reviewer nj1D both question the assumption that core and spurious features are spatially separable in the feature maps. In the rebuttal, we clarify that DAR is designed to exploit spatial separation when it exists. This is common in datasets where the foreground object contains the core features, while background context contains the spurious cues, and convolutional inductive biases tend to preserve the spatial separation in intermediate feature maps. Additionally, our other benchmark datasets are real-image datasets designed to study spurious correlations in more realistic settings where overlap between the spurious and core features can occur, and DAR still outperforms the other baselines. At the same time, we explicitly acknowledge, in our limitations section, that strict spatial separation does not hold universally and that DAR is not expected to provide additional gains in these scenarios. To make this limitation concrete, we added an ablation in Section 5.3.4 where the core and spurious signals fully overlapped; DAR no longer improves over existing methods in this regime but does not degrade performance either, clarifying the boundary of its applicability. We also emphasize that existing representation learning disentanglement techniques are complementary to DAR and can further enhance its effectiveness, as discussed in Section 6.
>
> 2. **DAR architectural design:**
>
> Reviewer NSbP raises a concern that the single-headed attention architecture would apply the same spatial attention weights across all feature maps, while reviewer nj1D questions the necessity of the proposed multi-head, two-layer attention architecture. In the rebuttal, we further support the architectural design choice via the added ablations in Section 5.3.1 (on Dominoes), comparing single-head vs multi-head and single-layer vs two-layer variants; the proposed architecture consistently outperforms these simpler alternatives, providing empirical support for the design choices and motivation outlined in Section 4.3.
>
> 3. **Confounding effect of introduced parameters:**
>
> Reviewer NSbP points out that the proposed attention module introduces extra trainable parameters and raises the concern that the observed improvements might stem from increased model capacity rather than from the specific architectural change. In the rebuttal, we first clarify the conceptual role of DAR: it explicitly exploits spatial structure to reweight core vs spurious regions before spatial collapse. This is studied in Section 4, where DAR shows its ability to extract a disentangled representation, which is not achievable by simply retraining more parameters in the DFR variants. We added a new ablation in Section 5.3.2, where we compare DAR with these DFR variants that introduce more trainable parameters than DAR, finding that they do not match DAR’s performance, which supports our claim that the improvement is due to the attention-based aggregation and its ability to selectively focus on informative spatial locations, rather than simply having an increased capacity.
>
> 4. **Dependence on representation quality:**
>
> Reviewer nj1D notes that, as a post-hoc method operating on a fixed ERM-trained backbone, DAR is inherently limited by the quality of the learned features. This is acknowledged in our limitations, and further explored in our ablation study in Section 5.3.3, where the backbone is first trained with a spectral decoupling loss designed to improve robustness under spurious correlations, and DAR is then applied on top of these features. This combination further improves performance, supporting the perspective that DAR is complementary to representation learning methods, as discussed in Section 6.
>
> [1] Yang, Y., Zhang, H., Katabi, D. and Ghassemi, M., 2023. Change is hard: A closer look at subpopulation shift. arXiv preprint arXiv:2302.12254.

---

### Meta-Review · Area_Chair_9UfU · 2026-01-04

**Summary:**

The paper studies the problem of debiasing models trained on data with spurious correlation. The authors extend the deep feature reweighting method (DFR), which retrains the last layer of a biased neural network on debiased data. The authors suggest that feature entanglement in the inputs to the last layer is a limitation for DFR, and replace the global average pooling layer with an adaptive attention layer, which can focus on parts of the input spatially. The authors show that the proposed Deep Attention Reweighting (DAR) improves over DFR on datasets where the core and spurious features are spatially separated.

The proposed method makes intuitive sense and is well-motivated. The authors conduct motivating analyses, and introduce novel metrics (CEP) for measuring the entanglement in representations. They also validate using visualizations (Fig 6) that the proposed method indeed can focus on the relevant part of the input better than DFR.

The core weakness of the paper is the strength of the empirical results:
- The method has limited applicability: it is designed for image data only, and it only works well when the core and spurious features are spatially distinct. This limits the number of datasets that the method can be applied to, and as a result the evaluation is only done on 3 datasets, all relatively toy (reviewers NSbP, w4eU). For example, the paper does not include CelebA (no spatial separation), MultiNLI and CivilComments (non-vision). The original DFR paper also included experiments on shape vs texture bias, which may also not be applicable as there isn't a spatial separation between core and spurious features. Admittedly, the authors perform additional experiments on the same data with different experimental configurations in Table 3. To summarize:
  + The applicability of the method is limited
  + The evaluation is limited
- Moreover, even on these datasets, the improvements compared to DFR are relatively small, in particular in the standard evaluation setting of Table 2 (reviewer w4eU).

To summarize, the paper has clear merits (clear motivation, good analysis), it is currently unclear if the method will have impact: in its current form, the method is only applicable to a narrow class of problems (vision problems with spatial separation between core and spurious features). Moreover, the empirical results do not strongly support the method even in this narrow setting, the improvements are small.

I believe the paper would be significantly strengthened by a broader evaluation, possibly identifying setting where the method provides a stronger improvement. At least, I believe it is important to evaluate the method on the standard datasets (at least in vision), even in settings where there is no spatial separation, to see if the method indeed doesn't help there (currently, only one such benchmark is reported as an ablation). You may also consider the ImageNet experiments from the original DFR paper, as those include datasets with background bias, where the method should be applicable.

**Reviewer Concerns:**

- Concerns on the limited applicability of the method (NSbP, nj1D): discussed, but not fully addressed
- Concerns on the limited evaluation (nj1D): discussed, but not fully addressed
- Concerns on lack of ablations (NSbP): at least partially addressed, authors added multiple new ablations
- Request for attention mask visualizations (NSbP): at least partially addressed, authors report GradCam visualizations
- Concern about the method being effective because of added parameters: addressed by ablations
- Concerns about the method reusing the same weights for all features: addressed, as multiple attention heads are used

**Reviewer Scores:**

- NSbP: 2 →  2/4
  + Concerns not fully addressed
- w4eU: 2 →  2
  + Responded to the original rebuttal maintaining their score; the authors have since responded again, but it seems unlikely that the reviewer would increase the score
- nj1D: 8 →  8
  + Responded to the rebuttal

---

### Decision · Program_Chairs · 2026-01-26

Reject